# Emotional Impact of COVID-19 Pandemic on Nursing Students Receiving Distance Learning: An Explorative Study

**DOI:** 10.3390/ijerph191710556

**Published:** 2022-08-24

**Authors:** Alfredo Manuli, Maria Grazia Maggio, Gianluca La Rosa, Vera Gregoli, Daniele Tripoli, Fausto Famà, Valentina Oddo, Giovanni Pioggia, Rocco Salvatore Calabrò

**Affiliations:** 1AOU Policlinico “G. Martino”, 98125 Messina, Italy; 2Department of Biomedical and Biotechnological Science, University of Catania, 95123 Catania, Italy; 3Department of Statistics, Università degli Studi di Messina-Piazza Pugliatti, 1, 98122 Messina, Italy; 4Oncology Unit, Azienda Ospedaliera Papardo, 98158 Messina, Italy; 5Institute for Biomedical Research and Innovation, National Research Council of Italy (IRIB-CNR), 98164 Messina, Italy; 6Neurorehabilitation Unit, IRCCS Centro Neurolesi “Bonino Pulejo”, 98121 Messina, Italy

**Keywords:** e-leaning, lockdown, nursing students, university

## Abstract

Social restrictions have a significant impact on higher education, especially on nursing students. The main goal of our study was to assess the emotional state of nursing students who received e-learning during the second wave of the COVID-19 pandemic. The secondary objective was instead to measure the usability and acceptability of distance learning systems. A cross-sectional survey design was used to assess the psychological effects of the COVID-19 pandemic on 1st-, 2nd-, and 3rd-year undergraduate nursing students attending the University of Messina, Italy, using an anonymous online questionnaire. The data of 522 nursing students were examined. All participants completed the online questionnaire, declaring the good usability of e-learning education (SUS mean 68.53 ds: 16.76). Moreover, we found that high levels of satisfaction in the use of the means of distance learning (based on the SUS score) were positively correlated with low levels of stress, anxiety, depression, and mental distress. In conclusion, the present study provided relevant information on usability and mental distress related to e-learning and use in a sample of nursing students. It was found that students generally found this method to be good for use. Although e-learning can be a valuable and usable teaching tool, the study suggests that students prefer a blended or presence modality, based on their perception of learning. So teaching nursing students in the future could integrate the two ways to enhance learning. Further studies are needed to evaluate this aspect.

## 1. Introduction

The COVID-19 pandemic is an infectious disease caused by the new coronavirus SARS-CoV-2. It is a global health problem that has affected millions of people since January 2020 [1]. The social restrictions resulting from COVID-19 had a personal and social impact, leading to the development of new models to cope with the global emergency. A significant change was recorded in the field of university education, especially in relation to the professions most involved in the emergency, such as nursing students [2]. It is known that nursing education must consider the technical skills and the emotional abilities to manage pathologies and stressful situations, especially when dealing with fragile patients [3]. Various authors have shown that nursing students have greater anxiety than other healthcare professionals, due to the challenges of the medical environment, the care of chronic patients, and the heaviness of the course and practical internships [4,5,6]. However, due to the pandemic, more than 1.5 billion students around the world have undergone the closure of schools and universities, with repercussions on university careers and personal growth, in terms of internships, subjective attendance, and reduced contact with professors and colleagues [4]. The closure of universities has forced institutions to use online training and digital e-learning tools, which have the advantage of being usable anywhere and at any time [7]. E-learning can be a useful and reliable tool for properly trained teachers to provide high-quality education [8,9]. However, the lack of face-to-face interaction could affect not only training but also increase stress, anxiety, and depression in students with a potential worsening of their learning skills [10,11,12]. In fact, university spaces and routines are important occasions for developing social networks, fostering peer support, and encouraging coping mechanisms [4]. University students may be stressed about changing habits and canceling educational events, such as experiences in foreign countries or graduation ceremonies [11,12,13]. Finally, the massive use of computerized equipment could lead to a sense of estrangement, encouraging ways of digital abuse, and students may not like such forms of teaching that could also be affected by technical problems [9,10]. Indeed, slow internet speed, power interruption, as well as the lack of face-to-face interaction have been perceived as barriers to satisfactory e-learning during the COVID-19 pandemic [13,14,15]. The satisfaction and usability of any technological tool used for distance learning are fundamental as it has been shown that the effectiveness of e-learning depends on optimism towards it and on the mastery of technologies [16]. Computer competence and usability of online tools would give students the opportunity to improve their skills [17,18], through digital literacy skills [19,20]. It was also found that in the case of good usability of online tools, this modality can stimulate learning, especially in the case of introverted subjects [21,22].

Thus, the present study had two objectives. The main goal was to assess the emotional state of a sample of college nursing students who received e-learning during the second wave of the COVID-19 pandemic. The secondary objective was to measure the usability and acceptability of distance learning systems.

## 2. Materials and Methods

### 2.1. Participants and Setting

We used a cross-sectional survey design. An anonymous online questionnaire was carried out using standard communication apps on smartphones (e.g., WhatsApp, Facebook) or via email. The 1st-, 2nd-, and 3rd-year university students of nursing who attend the University of Messina (Sicily, Italy) were involved. The final sample consisted of 521 students. The sociodemographic characteristics of the students are shown in Table 1.

All the students involved in the study used the same fully e-learning platform, “Microsoft Teams”, for their academic careers. Microsoft Teams (Microsoft 365) is a unified communication and collaboration platform that combines chat, conference/exam conference, content sharing (including contemporary work and file exchange), and application integration. It allows a quick material exchange and facilitates contact between teachers and colleagues, thus, allowing both the taking of exams and the attendance of academic lessons.

### 2.2. Procedures

Participants were interviewed online only, since it was not possible to use face-to-face modalities, due to the restrictive measures of the COVID-19 pandemic. They filled out the questionnaires in Italian through an online survey platform (“Google Form”, Google LLC, Mountain View, CA, USA) according to the CAWI (Computer-Assisted Web Interviewing) method. Data collection took place from October 2020 to March 2021. The research was carried out in the first semester of the courses, but after the first lockdown; thus, the students had already experienced e-learning.

The study was conducted according to the principles contained in the Declaration of Helsinki and approved by the local Ethics Committee (CE-ME.421/20). All students provided informed consent to participate in the research. In fact, at the beginning of the survey, the research was explained and participants were electronically asked to check “yes or no”, based on their willingness and consent to participate.

### 2.3. Study Development

The closed-ended questionnaire consisted of three sections: the first one collected information on the sociodemographic characteristics of each student; the second one contained the Depression Anxiety Stress Scales-21 (DASS-21); the third section concerned the System Usability Scale (SUS). In particular, the latter two areas refer to the clinical assessment tools used in this study. DASS-21 evaluates three constructs: depression, anxiety, and stress. It is a questionnaire of 21 questions on a 4-point Likert scale. Based on the score obtained, we can evaluate the degree of anxiety, depression, and stress that the subject showed during the administration of the test in reference to the previous 8–10 days [23].

The System Usability Scale (SUS) is a ten-element survey: each item is evaluated by a 5-point Likert scale [24]. It provides a comprehensive view of subjective assessments of the usability of the DAD system. DAD is the Italian acronym for Didattica A Distanza, i.e., distance learning carried out by electronic devices, also known as e-learning.

### 2.4. Statistical Analysis

The data of 521 nursing students were examined. Data were entered and analyzed using the Statistical Package for the Social Sciences (SPSS) version 16.0 (IBM Corp., Armonk, NY, USA).

For the first aim’s sake, the statistical analysis of students’ emotional states was divided into two steps.

In the first step, we performed a Principal Component Analysis (PCA) on the 21 DASS items to determine a combined measure of the three constructs of Stress, Anxiety, and Depression, named Mental Distress. Here, seven specific questionnaire items were assigned for each of the three DASS-21 constructs (see [8]).

In the second step, we performed four Ordinal Logistic Regressions (Ologit) on the Stress, Anxiety, Depression, and Mental Distress total scores by categorizing them into five levels. Then, we assessed if the gender, the academic year, the student’s age, and the SUS score (as regressors) can influence the levels of the scores mentioned above.

For the second aim, we ran an Analysis of Variance (ANOVA) test to verify if the usability and acceptability of distance learning systems were perceived differently among the three academic courses, according to the corresponding year.

#### 2.4.1. Mental Distress Index and Analysis of Emotional Impact on the Nursing Students

In order to understand the global emotional impact of students, we combined two analyses. In the first step, we defined a combined index of the perceived level of Stress, Anxiety, and Depression (named Mental Distress) by using the Principal Component Analysis (PCA). In the second step, we performed an Ordinal Logistic Regression to identify which factors are associated with different levels of Stress, Anxiety, Depression, and Mental Distress.

##### Principal Component Analysis (PCA)

PCA is an exploratory multivariate analysis used to examine the overall structure of DASS-21 items. PCA allowed us to define a composite index to summarize adequately the scores of the three DASS-21 dimensions [10].

With this outlook, we defined the Mental Distress Score as an overall score of the students’ online education pressure. Specifically, Mental Distress is defined as the (negative) students’ state of mind about the impossibility of reaching University, the interruption of their social relationships, and the changes in their learning habits due to the pandemic restrictions.

The reasoning concerning this combined construct stems from the idea of defining a more general factor able to simultaneously capture the coexistence of all the three DASS-21 dimensions (i.e., Depression, Anxiety, and Stress) [8].

Additionally, we ran Cronbach’s alpha (α) to evaluate the reliability of the DASS-21 questionnaire [10]. Cronbach’s alpha (α) measures internal consistency across individual indicators. In this vein, the αs for the four constructs range from 0.885 to 0.961: all of the four coefficients were over the 0.70 thresholds, showing excellent internal consistency of the three scales of the DASS-21 and the combined overall score.

##### Ordinal Logistic Regression (Ologit)

Ologit extends the general linear model to ordinal categorical data [11]. In Ologit, the event of interest is to observe a score where j = 1, 2, …, M − 1. M is the number of categories of the ordinal response variable. So, in our case, the categories we will analyze will be M-1 = 4. For this reason, and according to the Ologit model, four constants () for each regression had been estimated [11]. The Ologit equation can be written as
P (Yi > j) = g(Xβ) = exp (αj + Xiβ)/1 + {exp (αj + Xiβ)}

Here, Stress, Anxiety, Depression, and Mental Distress scores were considered as ordinal dependent variables, where each specific level was coded as 1-normal, 2-mild, 3-moderate, 4-severe, 5-extremely severe. The independent variables identified were: gender (coded as 1 = male, 0 = female), academic year (coded as 1 = first year, 2 = second year, 3 = third year), the student’s age, and SUS score (defined as discrete variable).

## 3. Results

Although we contacted 1150 students, the final sample consisted of 521, with a mean age of 21.52 ± 3.28, (128 males—24.6%; 393 females—75.4%); 340 students (65.26%) belonged to the 1st year; 120 students (23.03%) of the 2nd year and 61 students (11.70%) of the 3rd academic year. Thus, about 70% of the online contacted subjects did not respond. This can be due both to emailing problems that can create missing delivery (the email can end up in spam, the message cannot be read), or to personal reasons, such as boredom in compiling, or reluctance to give consent to use data for the research. Figure 1 and Figure 2 summarize the most relevant results.

### 3.1. Statistical Results on Emotional State

To study the relationships between the four scores and the defined regressors, four ordinal logistic regressions were run. Among the shared results, high levels of satisfaction in the use of distance learning tools (based on the SUS score) lead to low levels of Stress, Anxiety, Depression, and Mental Distress, emphasizing the importance of e-learning in managing the restrictions due to the COVID-19 emergency and ensuring the continuity of academic practices. As shown in Table 2, the relationships between the four DASS-21 scores and student gender and age were significant: younger students experienced higher levels of negative emotions related to the four categories. From an academic perspective, female students dealt with the lockdown restrictions better than males concerning the Anxiety and Depression aspects. Lastly, no significant relationship was found between the academic year and the four proposed DASS-21 scores: belonging to a specific year did not influence any scores.

### 3.2. Usability Results

All participants who gave their consent completed the entire online questionnaire, declaring the good usability of e-learning education (SUS mean 68.53 SD: 16.76). Furthermore, 48% of the students were pleased to use the DAD, 43.30% expressed a negative opinion about its use, while 8.7% considered the tool neither useful nor unuseful. As for the possible use of e-learning in the future, 37.55% stated that they would recommend the tool also post-pandemic, 56.90% would not want to use it, while 5.60% expressed that they do not know if they would like to continue using it. Finally, almost all of the sample (88.13%) declared that e-learning is not a tool that can be compared to face-to-face lessons. Considering the normality of SUS data assessed by the Jarque–Bera test, a one-way ANOVA has been performed on the SUS score. As described in Table 3, we found no statistical significance between the means of the three academic groups concerning the SUS score (F = 0.55, *p*-value = 0.58). Here, the students of the first, second, and third years perceived the usability and acceptability of distance learning systems in a similar way.

## 4. Discussion

The aim of our study was to evaluate the emotional repercussions of e-learning on nursing students belonging to an Italian University and the usability of the platform to attend online lessons. Although this latter was a secondary aim, we found that high satisfaction with the use of e-learning (based on the SUS score) leads to low levels of stress, anxiety, depression, and mental distress. This further demonstrates, in our opinion, that usability could influence the perception of e-learning in terms of mental distress and mood.

As it is known, the relationship between learning, stress, and mood is an essential component in the development and growth of students. An increase in negative mood and stress can hinder learning and the acquisition of new information, due to the close relationship between emotions, cognitive processes, and academic performance [25,26,27]. In particular, it is important to examine the relationship between mental distress and mood in students, and then also relate them to usability for a variety of reasons. First of all, the perception of anxiety and stress and the sense of low responsibility in learning negatively affect the cognitive components and academic progress [25,28,29,30]. Although this latter aspect was not directly assessed in our study, previous research has pointed out that stress [31,32], depression [33,34], and anxiety [34,35] are risk factors for poor academic achievement or even failure. Therefore, it is essential to evaluate the factors that can counteract these emotional symptoms and positively impact learning. Alqurashi et al. found that a satisfying experience with computer usability was positively and significantly associated with adequate learning, higher satisfaction and lower depression or stress [36], as also shown by other authors [37,38], especially in studies carried out on nursing students [39,40]. Kim et al. found that nursing students declared that e-learning had an impact on their empowerment and the feeling of being in control of their learning situation: this can counteract mental stress and promote an improvement in the mood [40]. Moreover, thanks to adequate technological support, students can recover lessons from place and time, reducing the dispersion and stress of traveling to the facility, especially if students come from distant places [41,42]. Our data confirm the findings of recent research that e-learning is considered a positive aid for nursing students [40,43,44]. In fact, Unstead et al. [43] found that if the online tools have good usability, e-learning seems feasible and useful in training nursing students. Another interesting study was carried out by Herbert et al. [45]. The authors evaluated, in a sample of nursing students, the technology acceptance and usability of the same online software platform used for e-learning also by our sample. Participants perceived Microsoft Teams as a useful and easy-to-use tool that has positively supported the teaching–learning process [45].

Furthermore, a digitized approach to learning was highly valued [46], and many students believe that e-learning can increase a sense of personal responsibility and control over their learning [47].

Despite these encouraging results, other authors have shown students’ negative attitudes toward e-learning, due to reduced socialization and a lack of face-to-face internship experiences with colleagues, teachers, patients, and hospital settings [48,49,50,51]. This aspect is central to nursing education, as it involves multiple practical sessions, both in terms of laboratory and bedside clinical training. For this reason, some authors underline that the lack of contact affects the attitudes related to e-learning [48,49,50,51]. This important issue requires investment in infrastructure and teaching skills development to make students’ learning satisfactory [52,53]. In line with these studies, although e-learning is considered easily usable and a valid educational alternative, our data showed that students did not have a positive attitude towards these tools, especially for their use in the future. The participants, in fact, declared that they preferred face-to-face lessons or in mixed modality, which combines the advantages of online theoretical sessions with clinical practice in the hospital or at the patient’s bed.

Since attitude toward e-learning is fundamental, especially for nursing students, this construct should be investigated by using specific and validated scales, such as the 29-item NSA-E-learning scale (which was found valid and reliable in a Turkish student population) [54].

An interesting finding of our study is the significant relationship between sex and age of participants with mental distress: younger students felt more uncomfortable with the pandemic restrictions with respect to their older colleagues, whereas, regarding anxiety and depression and gender, it was observed that female students dealt with the pandemic better than their male peers. The results are in agreement with Yekefallah et al. [50], who found that female students tend to perform better, even online, than male students. Although these issues have also been confirmed by other studies [55,56], some authors have found no gender differences in students’ mental distress [57,58]. In particular, recent research by Kabir et al. [59] found that e-learning readiness had a significant association with perceived e-learning stress among university students. Moreover, different variables including age, gender, parents’ highest education level, devices used as well as having a single room were the potential predictors of e-learning readiness.

On the other hand, it has been found that there was no significant increase in dissatisfaction and stress as well as no gender differences with e-learning, since nursing students declared that, even with the constraints, they will become good and competent practitioners based on their initiative and will. In fact, the underlying knowledge they can gain from the course will prepare them to retrieve such information also in clinical practice [60]. According to Yekefallah et al. [50], we believe that this discrepancy is due to the different populations used as a sample and to the non-homogeneity of the assessment tools used.

Professional identity, which refers to the positive perception, evaluation, and emotional experience of the nursing profession and identity to be undertaken, could be considered another factor potentially affecting adaptability to e-learning [61].

This issue deserves future investigation, also by evaluating its potential correlation with stress and mood disorders.

Another poorly investigated issue from our study is the age difference in the perception of emotional distress. Our data are confirmed by Narimani et al. [62], who showed that increasing age is correlated with a lower level of distress. However, in our sample, there are no significant differences related to the academic year attended, but further research will be useful to evaluate the differences in discomfort related to the academic year attended.

Finally, a novelty of our study is the definition of the Mental Distress index starting from the total score of the DASS-21, which includes the three scales of anxiety, depression, and stress. Further validation studies are necessary to ascertain the adequacy of this construct.

Despite the interesting results, our study has some limitations. First, the survey was submitted via online forms. This mode, required by the lockdown, did not allow for both monitoring of the participants during the compilation of the questionnaires and supervision of any disturbing elements (such as distraction). Furthermore, the test user does not present the index of the mental distress scale as validated in the Italian context.

Further research on this important topic is needed to confirm our findings and help find new approaches to better deal with the educational issues during pandemics.

## 5. Conclusions

In conclusion, the present study provides relevant information on e-learning usability and the related mental distress in a sample of nursing students. It was found that students generally found this method useful, and the high levels of usability seem to correlate with a reduction in mental distress and better mood. However, although e-learning can be a valuable and usable teaching tool, the study suggests that students prefer a blended or presence modality, based on their perception of learning. Then, teaching methods for nursing students in the future could integrate the two ways to enhance learning and reduce mental stress and related problems.

## Figures and Tables

**Figure 1 ijerph-19-10556-f001:**
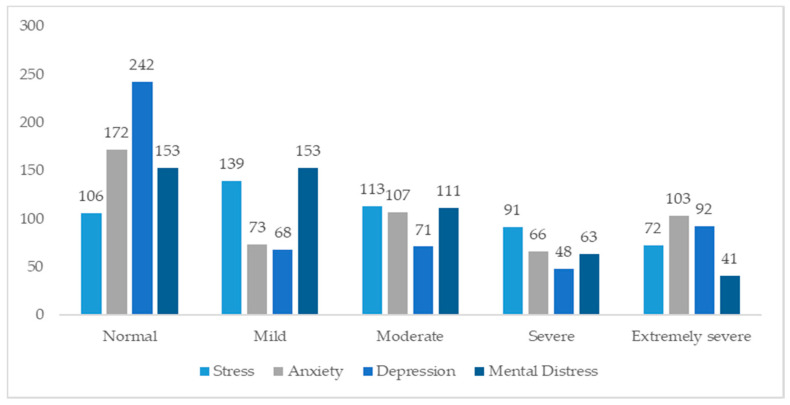
Classification of nursing students (n = 521) according to their levels of Stress, Anxiety, Depression, and Mental Distress.

**Figure 2 ijerph-19-10556-f002:**
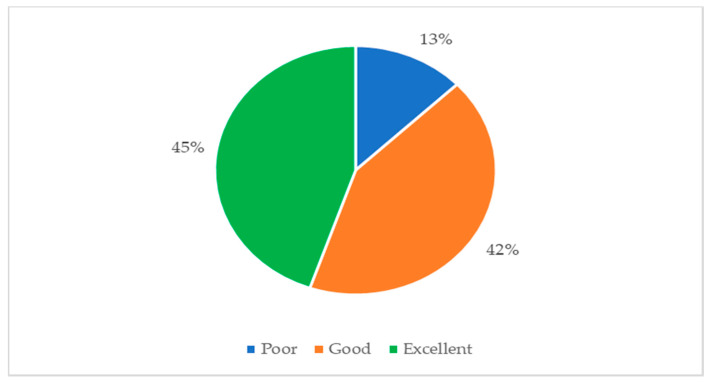
Classification of nursing students (n = 521) according to their SUS score.

**Table 1 ijerph-19-10556-t001:** Sociodemographic and characteristics of the students (n = 521).

Sociodemographic Variables	Value
Gender	
Female	128 (24.57%)
Male	393 (75.43%)
Age (years)	21.52 ± 3.28
Education (years)	14.36 ± 2.73
Academic year	
First	340 (65.26%)
Second	120 (23.03%)
Third	61 (11.71%)

Mean ± standard deviations were used to describe continuous variables; proportions (%) were used to describe categorical variables.

**Table 2 ijerph-19-10556-t002:** Ordered logistic regression by DASS-21 scores (beta estimates).

	Stress	Anxiety	Depression	Mental Distress
Variable	Coefficient	Coefficient	Coefficient	Coefficient
Gender	1.018 ***	0.759 ***	0.998 **	1.029 ***
Academic year	0.054	0.105	0.083	0.106
Age	−0.046 *	−0.060 ***	−0.097 ***	−0.061 **
SUS	−0.040 ***	−0.035 ***	−0.039 ***	−0.039 ***
Constant (1)	−4.399 ***	−3.739 ***	−4.015 ***	−4.043 ***
Constant (2)	−2.953 ***	−3.077 ***	−3.392 ***	−2.621 ***
Constant (3)	−1.935 ***	−2.127 ***	−2.685 ***	−1.469 **
Constant (4)	−0.801	−1.400 ***	−2.088 ***	−0.315
N	521	521	521	521
LR χ^2^	101.33 ***	78.73 ***	100.18 ***	103.18 ***

* Significance levels of 10% (*), 5% (**), and 1% (***) for coefficients by z-test and likelihood-ratio χ^2^ test.

**Table 3 ijerph-19-10556-t003:** One-way ANOVA on SUS score.

Source of Variance	SS	df	MS	F	*p*-Value
Between groups	310.91	2	155.40	0.55	0.576
Within groups	145,928.32	518	281.71		
Total	142,339.23	520	281.23		

Note: SS (Sum of Square), df (degrees of freedom), MS (Mean Square).

## Data Availability

On demand to the corresponding author.

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
