# Peer review of "Emotional Impact of COVID-19 Pandemic on Nursing Students Receiving Distance Learning: An Explorative Study"

_ijerph, 2022, doi:10.3390/ijerph191710556_

Round 1

Reviewer 1 Report

The topic of the article is current; finding out the impacts of the COVID pandemic and the changes they have caused in the field of information technology and human health is necessary. However, the overall goal of the article seems a bit inconsistent - on the one hand, the emotional status of a student sample university is examined, and on the other hand, the usability and acceptability of the remote learning system are inspected. The authors try to connect these two types of data and look for a connection between them.

At first glance, it is a connection of two different topics between which there may be an apparent connection, but in the age of covid, the use of e-learning was not the reason that caused mental distress. I assume that the questions from DAS21 were used in the original wording. If the authors modified them concerning virtual education, it was necessary to state these facts.

If the article should cover both topics, it is necessary to justify their connection or relationship.

In the related work section, there is no review of works that deal with similar issues. I have already reviewed several of them, so it is impossible to say that the topic is new, but it is necessary to look at the issue from several points of view. The authors refer to several works in the conclusion to confirm the results, but within the related work, they could also mention the methods and procedures used and inspire them.

I positively assess that two standardized questionnaires - DASS21 and SUS - were used in the study.

There is no explanation why Principal Component Analysis was used in chapter 2.4. Furthermore, the statement "This approach allowed us to identify a latent dimension by analyzing the unknown structure of the data" deviates from the concept of using standardized questionnaires and the fact that the evaluation of questionnaire results also represents a standardized process.

What does Fig. 1 represent? Is it about including questions in the questionnaire based on a standardized structure or about an author's proposal (if it is the second option, you must justify why)?

Chapter 2.4 would deserve an overall better explanation of the choice of methods and also the purpose of their application. However, I still consider the connection between SUS and DAS21 inappropriate in the context of COVID.

The authors refer to the Mental Distress Index in several places, but the article lacks its introduction, explanation, and justification of its correctness. It is only very superficially mentioned in the first paragraph of part 3.2.

The meaning of table 2 is also not clear. For example, what is the purpose of determining the correlation between individual items of questionnaires?

The content of table 3 and the values presented in it must also be described.

Cutpoints in table 4 need to be described and explained, what they express and why they are important?

The scope of the conclusion is tiny; I recommend combining it with the discussion.

It is not appropriate to publish the article in this form; it is desirable to expand it with the facts mentioned above and procedures and possibly supplement it with graphs to visualize the obtained results.

Author Response

The topic of the article is current; finding out the impacts of the COVID pandemic and the changes they have caused in the field of information technology and human health is necessary. However, the overall goal of the article seems a bit inconsistent - on the one hand, the emotional status of a student sample university is examined, and on the other hand, the usability and acceptability of the remote learning system are inspected. The authors try to connect these two types of data and look for a connection between them.

Thanks for the comment, we have proceeded to clarify. There are two goals of the study and we have revised the paper to look for a connection between them. We have also changed the results order to better fit with aim and discussion

At first glance, it is a connection of two different topics between which there may be an apparent connection, but in the age of covid, the use of e-learning was not the reason that caused mental distress. I assume that the questions from DASS21 were used in the original wording. If the authors modified them concerning virtual education, it was necessary to state these facts. If the article should cover both topics, it is necessary to justify their connection or relationship.

The DASS-21 was used in the standardized version of the Italian sample (for further information: Bottesi, G.; Ghisi, M.; Altoè, G.; Conforti, E.; Melli, G.; Sica, C. The Italian version of the Depression Anxiety Stress Scales-21: Factor structure and psychometric properties on community and clinical samples. Compr Psychiatry 2015, 60, 170-181). In addition to the three recognized indices in the version, we also added a global index, validating it by the analysis. We have put the part of the DASS-21 that may seem off-topic in the supplementary file. We have two goals in the study, we looked at both aims.

In the related work section, there is no review of works that deal with similar issues. I have already reviewed several of them, so it is impossible to say that the topic is new, but it is necessary to look at the issue from several points of view. The authors refer to several works in the conclusion to confirm the results, but within the related work, they could also mention the methods and procedures used and inspire them.

The rater is right with this issue. Besides some cited papers, other studies on different populations in relation to e-learning by our group (i.e. Maggio MG, Stagnitti MC, Calatozzo P, et al. What about the Consequences of the Use of Distance Learning during the COVID-19 Pandemic? A Survey on the Psychological Effects in Both Children and Parents. Int J Environ Res Public Health. 2021;18(23):12641. Published 2021 Nov 30. doi:10.3390/ijerph182312641) has been recently published.  Our curiosity was also to define these aspects in a population of nursing students because they are more exposed to the pandemic emergency.

I positively assess that two standardized questionnaires - DASS21 and SUS - were used in the study.

Thanks for the comment.

There is no explanation why Principal Component Analysis was used in chapter 2.4. Furthermore, the statement "This approach allowed us to identify a latent dimension by analyzing the unknown structure of the data" deviates from the concept of using standardized questionnaires and the fact that the evaluation of questionnaire results also represents a standardized process.

We added an explanation in Section 2.4 of its importance in determining the combined index of the three dimensions of DASS (i.e., Anxiety, Depression, and Stress), named Mental Distress.

What does Fig. 1 represent? Is it about including questions in the questionnaire based on a standardized structure or about an author's proposal (if it is the second option, you must justify why)?

Fig. 1 represents the way in which PCA works on the development of the combined Mental Distress index. We removed this figure to make the card clearer, as it was out of the aims.

Chapter 2.4 would deserve an overall better explanation of the choice of methods and also the purpose of their application. However, I still consider the connection between SUS and DAS21 inappropriate in the context of COVID.

Thanks for the comment, we have proceeded to clarify.

The authors refer to the Mental Distress Index in several places, but the article lacks its introduction, explanation, and justification of its correctness. It is only very superficially mentioned in the first paragraph of part 3.2.

We have added the missing information, as suggested.

The meaning of table 2 is also not clear. For example, what is the purpose of determining the correlation between individual items of questionnaires?

We have proceeded to eliminate it for a better understanding of our work.

The content of table 3 and the values presented in it must also be described.

Thanks for the comment. We have modified the table accordingly.

Cutpoints in table 4 need to be described and explained, what they express and why they are important?

Cutpoint is the alternative name for Constant in the OLOGIT regression. This model feature has been highlighted in 2.4 paragraphs.

The scope of the conclusion is tiny; I recommend combining it with the discussion.

We have made the required changes.

It is not appropriate to publish the article in this form; it should be expanded with the facts and procedures mentioned above and possibly supplemented with graphs to view the results obtained.

We have modified the work as required.

Reviewer 2 Report

Dear auhtors,

I understand that it is a valuable study. But I felt there was a problem with the presentation. Also, the conclusions did not seem to be supported by statistical analysis. Theoretically, it is understandable that the usability of the system reduces stress. However, I don't think there is enough evidence that the statistical results show that.

table 1.

  Since the title of the table says "student", "student 100%" in the table seems unnecessary.

  The annotations on the table are a little confusing. Please review including the grammar.

”Proportions (numbers and percentages) were used to describe categorical variables”

  I feel this explanation is not necessary.

2.1.Participants and settings 

 The title is participants and settings, but I think there is only information about the participants.

2.2. Procedure

"Participants were interviewed online"

  Indeed, this research is being conducted using Computer Assisted Web "Interviewing". However, readers might think meeting at which someone is asked questions in order to find out something when they read "interview". 

I think it is necessary to describe whether it was a completely online education or whether it was partly face-to-face.

There are also various types of online education. Authors need to explain what kind of online education the students received and what percentage they received.

"The study complies with the principles contained in the Helsinki Declaration, and all participants provided informed consent to participate"

  There is no period.

  Also, I think the authors should explain how inform to participants and how to recieve their consent.

2.3. Study development

What kind of period for the students did the authors take place the survey? Is it just the beginning of the semester, or is it the time when you have had enough student life? Or was it the test period?

"the third section concerns the System Usability Scale (SUS)."

  System Usability Scale of what?

"The Depression Anxiety Stress Scales (DASS-21) evaluates three constructs: depression, anxiety, and stress."

  It's the second time, so I think "DSAA-21" is fine. Spelling out is not neccesary.

"The questionnaire measures depression, anxiety, and stress, which are-"

  It is not necessary because the explanation in the previous sentence is repeated.  

"The descriptive statistics were analyzed and expressed as mean ± standard deviation or as median ± first third quartile for continuous variables, as appropriate; frequencies (%) were used for categorical variables.The perception of usability of the questionnaire was expressed in percentages."

  I don't feel the need for this explanation. This is because the calculation of the mean, standard deviation, and percentage is the descriptive statistics themselves.

"Principal Component Analysis (PCA)" is repeated.

Figure 1.

  Items 1 to 21 are described in the figure. I don't think this figure makes sense if it's not clear what each item represents.

Also, I think this is the result of principal component analysis, not the method.

P (Yi > j) = g(Xβ) = exp⁡(j + Xiβ)1 + {exp⁡(j + Xiβ)}

  Are all variables described in the text? Also, please check the validity of the formula.

340 students (65.26%) belonged to the 1st academic year; 120 students (23.03%) of the 2nd Academic Year and 61 students (11.70%) of the 3rd Academic Year.

  "Academic Year" is repeated. Also, "academic" and "Academic" are mixed.

"All participants completed the online questionnaire-"

  Is it true? It seems unlikely that everyone answer. The authors should explain in method section how you did to participant before survey. Also, it would be better to explain why the respons rate was 100%.

"(SUS mean 68.53 ds: 16.76)."

ds is sd?

There is no explanation as to what kind of system the students are using, so I feel that it makes no sense to show only this mean score. I think you should explain the system used by students somewhere. Students use only one system? If there are more than one, which system do students answer for SUS?

3.1. Usability Results

  Is there a difference in the SUS score depending on the grade and gender? How about doing a crosstab to create a table?

3.2. PCA Results

  Because the variable nemes used in PCA, the presentation in Table 2 seems meaningless.

"With this outlook, we defined the Mental Distress Score as an overall score of the students' lockdown experience and their relationship with COVID-19."

  This study analyzes the stress of online education. I feel that the expression "lockdown experience" is a leap forward, and I think it deviates from the purpose of this research.

In this vein, the αs for the four constructs range from 0.085 to 0.961: all of the four coefficients were over the 0.70 thresholds, showing excellent internal consistency of the three scales of the DASS-21 and the combined overall score (table 3).

  Only the correlation coefficient is written in Table 3.

  Since stress, anxiety, and depression are components of mental stress, are their correlation coefficients meaningful?

"Significance levels of 1% are in bold"

  Those that were significant on a 1% level are in bold. 

"Preliminarily, table 4 shows the correlation between the four DASS-21-related quantitative scores (Stress, Anxiety, Depression, and Mental Distress) and the SUS score."

  Table 3 ??

3.3. Ologit Results

"Among the shared results, high levels of satisfaction in the use of the means of distance learning (based on the SUS score) lead to low levels of Stress, Anxiety, Depression, and Mental Distress, emphasizing the importance of e-learning in managing the restrictions due to the COVID-19 emergency and ensuring the continuity of academic practices."

  Is this a judgment from the partial regression coefficient in Table 4? The coefficient is extremely small, and I don't think it makes any sense. The same is true for age. The coefficient is very small. Also, the range of ages of the subjects is small. Will this small age difference actually make a practical difference in how stress is felt?

  There is no explanation for the cutpoint.

4. Discussion

What is written in the discussion seems reasonable at first glance. However, as I have pointed out, I do not think that the content of this discussion is dictated by the statistical analysis of this study. Therefore, no explanation can be judged to be valid.

5. Conclusions

"In conclusion, although e-learning can be a valid and usable teaching tool, our study highlighted that face-to-face teaching is essential for nursing students because it allows internships and practical experiences necessary for the education of future healthcare professionals."

  The study aims to evaluate the emotional status of a sample of university nursing students who received e-learning. I feel that the conclusion is not suitable for the purpose.

"The participants, in fact, declared that they preferred face-to-face lessons or in mixed modality, which combines the advantages of online theoretical sessions with clinical practice in the hospital or at the patient's bed."

  The conclusions of this study appear to be based here. If so, there is not much point in conducting statistical analysis, and it is sufficient to simply listen to the opinions of students and reflect them in the education system.

"Furthermore, the test user does not present the mental distress scale as validated in the Italian context, so further studies will be needed to confirm our results.."

  There are two periods.

Author Response

I understand that it is a valuable study. But I felt there was a problem with the presentation. Also, the conclusions did not seem to be supported by statistical analysis. Theoretically, it is understandable that the usability of the system reduces stress. However, I don't think there is enough evidence that the statistical results show that.

We modified the study to make the work more coherent and adapted the statistical analysis to the objectives.

table 1.

Since the title of the table says "student", "student 100%" in the table seems unnecessary.   The annotations on the table are a little confusing. Please review including the grammar.

”Proportions (numbers and percentages) were used to describe categorical variables”.   I feel this explanation is not necessary.

Done.

2.1.Participants and settings

 The title is participants and settings, but I think there is only information about the participants.

We have changed the title.

2.2. Procedure

"Participants were interviewed online"

Indeed, this research is being conducted using Computer Assisted Web "Interviewing". However, readers might think meeting at which someone is asked questions in order to find out something when they read "interview".

Done.

I think it is necessary to describe whether it was a completely online education or whether it was partly face-to-face. There are also various types of online education. Authors need to explain what kind of online education the students received and what percentage they received.

The final sample consisted of 521 students who implemented the training completely online, using e-learning modalities. We have included this in the “participants” section.

"The study complies with the principles contained in the Helsinki Declaration, and all participants provided informed consent to participate".  There is no period.   Also, I think the authors should explain how to inform participants and how to receive their consent.

We corrected the mistake and added the information, as requested.

2.3. Study development

What kind of period for the students did the authors take place the survey? Is it just the beginning of the semester, or is it the time when you have had enough student life? Or was it the test period?

The research was carried out in the first semester of the courses (from October 2020 to March 2021), but after the first lockdown, thus, the students had already experienced e-learning.

"the third section concerns the System Usability Scale (SUS)."  System Usability Scale of what?

Usability is related to e-learning, as we described in the following paragraph:

“The System Usability Scale (SUS) is a ten-element survey: each item is evaluated by a 5-point Likert scale [9]. It provides a comprehensive view of subjective assessments of the usability of the DAD system. DAD is the Italian acronym for Didattica A Distanza, i.e., distance learning carried out by electronic devices, also known as e-learning.”

"The Depression Anxiety Stress Scales (DASS-21) evaluates three constructs: depression, anxiety, and stress."  It's the second time, so I think "DSAA-21" is fine. Spelling out is not necessary.

Done.

"The questionnaire measures depression, anxiety, and stress, which are-"   It is not necessary because the explanation in the previous sentence is repeated. 

Done.

"The descriptive statistics were analyzed and expressed as mean ± standard deviation or as median ± first third quartile for continuous variables, as appropriate; frequencies (%) were used for categorical variables. The perception of usability of the questionnaire was expressed in percentages." I don't feel the need for this explanation. This is because the calculation of the mean, standard deviation, and percentage is the descriptive statistics themselves.

"Principal Component Analysis (PCA)" is repeated.

Done.

Figure 1.   Items 1 to 21 are described in the figure. I don't think this figure makes sense if it's not clear what each item represents.

Thanks for the comment. We have proceeded to eliminate it to reach a better understanding of our work.

Also, I think this is the result of principal component analysis, not the method.

P (Yi > j) = g(Xβ) = exp⁡(j + Xiβ)1 + {exp⁡(j + Xiβ)}

There was a formatting error in the definition of the OLOGIT formula. This error has been corrected.

  Are all variables described in the text? Also, please check the validity of the formula.

All variables used in the model were mentioned. We checked the validity of OLOGIT equation

340 students (65.26%) belonged to the 1st academic year; 120 students (23.03%) of the 2nd Academic Year and 61 students (11.70%) of the 3rd Academic Year.   "Academic Year" is repeated. Also, "academic" and "Academic" are mixed.

Done.

 "All participants completed the online questionnaire-"   Is it true? It seems unlikely that everyone answer. The authors should explain in method section how you did to participant before survey. Also, it would be better to explain why the respons rate was 100%.

We mean all 522 who have agreed to participate and have given consent.

In fact, in the previous paragraph we wrote:

"Although we contacted 1150 students, the final sample consisted of 522 students with a mean age of 21.52 ± 3.28, (128 males -24.6%; 392 females - 75.4%); 340 students ( 65.26%) belonged to the 1st year; 120 students (23.03%) of the 2nd year and 61 students (11.70%) of the 3rd academic year. "

We have clarified the concept.

"(SUS mean 68.53 ds: 16.76)."

ds is sd?

We corrected the mistake.

There is no explanation as to what kind of system the students are using, so I feel that it makes no sense to show only this mean score. I think you should explain the system used by students somewhere. Students use only one system? If there are more than one, which system do students answer for SUS?

All the students involved in the study used the same e-learning platform, called "Moodle UNIME". The platform offered numerous services to students, such as enrollment in courses, news, and updates from the reference teacher, student forum, video lessons, and exam results. We have explained this aspect in the “participants” section.

3.1. Usability Results

  Is there a difference in the SUS score depending on the grade and gender? How about doing a crosstab to create a table?

We run a one-way ANOVA test to verify if there are significant differences in SUS scores among the three academic courses.

3.2. PCA Results

  Because the variable nemes used in PCA, the presentation in Table 2 seems meaningless.

We have proceeded to eliminate this aspect, as suggested.

"With this outlook, we defined the Mental Distress Score as an overall score of the students' lockdown experience and their relationship with COVID-19." This study analyzes the stress of online education. I feel that the expression "lockdown experience" is a leap forward, and I think it deviates from the purpose of this research.

We have proceeded to modify to better understand our work.

In this vein, the αs for the four constructs range from 0.085 to 0.961: all of the four coefficients were over the 0.70 thresholds, showing excellent internal consistency of the three scales of the DASS-21 and the combined overall score (table 3).   Only the correlation coefficient is written in Table 3.

We have eliminated it, as suggested.

  Since stress, anxiety, and depression are components of mental stress, are their correlation coefficients meaningful?

 "Significance levels of 1% are in bold"

  Those that were significant on a 1% level are in bold.

We removed this aspect, as suggested.

"Preliminarily, table 4 shows the correlation between the four DASS-21-related quantitative scores (Stress, Anxiety, Depression, and Mental Distress) and the SUS score."

We have proceeded to eliminate it to reach a better understanding of our work.

  Table 3 ??

Thanks for the comment. We have proceeded to correct it.

 3.3. Ologit Results

"Among the shared results, high levels of satisfaction in the use of the means of distance learning (based on the SUS score) lead to low levels of Stress, Anxiety, Depression, and Mental Distress, emphasizing the importance of e-learning in managing the restrictions due to the COVID-19 emergency and ensuring the continuity of academic practices."   Is this a judgment from the partial regression coefficient in Table 4? The coefficient is extremely small, and I don't think it makes any sense. The same is true for age. The coefficient is very small. Also, the range of ages of the subjects is small. Will this small age difference actually make a practical difference in how stress is felt?

We rewrote this section to clarify the analysis and to make it in line with the aims.

 There is no explanation for the cutpoint.

Cutpoint is the alternative name for Constant in the OLOGIT regression. This model feature has been highlighted in 2.4 paragraphs.

  1. Discussion

What is written in the discussion seems reasonable at first glance. However, as I have pointed out, I do not think that the content of this discussion is dictated by the statistical analysis of this study. Therefore, no explanation can be judged to be valid.

We revised the text to make discussion more reasonable.

  1. Conclusions

"In conclusion, although e-learning can be a valid and usable teaching tool, our study highlighted that face-to-face teaching is essential for nursing students because it allows internships and practical experiences necessary for the education of future healthcare professionals."

We removed this consideration.

  The study aims to evaluate the emotional status of a sample of university nursing students who received e-learning. I feel that the conclusion is not suitable for the purpose. "The participants, in fact, declared that they preferred face-to-face lessons or in mixed modality, which combines the advantages of online theoretical sessions with clinical practice in the hospital or at the patient's bed." The conclusions of this study appear to be based here. If so, there is not much point in conducting statistical analysis, and it is sufficient to simply listen to the opinions of students and reflect them in the education system.  "Furthermore, the test user does not present the mental distress scale as validated in the Italian context, so further studies will be needed to confirm our results.."   There are two periods.

We revised the text.

Reviewer 3 Report

Dear author,

thank you for the opportunity to read your manuscript. Please see attached the comments in the PDF file.

Author Response

Thanks for the suggestions, we have made the changes and added new references.

Reviewer 4 Report

Dear authors

The theme of this study is very interesting.

Manuscript attached with comments

kind regards

Author Response

Thanks for the suggestions, we have made the required changes

Round 2

Reviewer 1 Report

Unfortunately, I consider the submitted version of the article to be inconsistent in terms of formality and format.

The added text in section 2.1 is messy, and in the last paragraph in section 1 again. This situation is probably the result of tracking changes, but it is advisable to check the author's reviewed document before submitting it.

A description of what Microsoft Teams is probably not the appropriate content for this type of article. Moreover, this information is not at all necessary for the set goals.

The added content in section 2.4 is again chaotic; on page 7 again.

The used form of change tracking saved into pdf format is very confusing. It is more appropriate to use docx format or color differentiation of changed parts without bubbles in the future.

Statement on the incorporation of comments from the previous review round:

I agree with the definition of the two goals of the study in the revised paper. However, I still do not see the relationship between these goals. On what basis do the authors conclude that there is a relationship between these characteristics? Why did they decide to investigate the correlation between them? What should the finding of correlation bring?

If at least one of these questions cannot be answered, the article in this form makes no sense, and it is appropriate to divide it into two.

The common goal is partially touched by the definition of Mental Distress in section 2.4. Still, it is the GOAL of the article, and it is necessary to define it or its starting points or the idea of a solution in a clear form earlier.

Several of my comments were related to the used bibliographical sources. Although I received several responses to the comments from the authors, they mainly did not satisfy me. In addition to the author team, other authors are undoubtedly dealing with related topics. An analysis of their results and approaches would enrich the article in Related work, which is not as represented in the article as I would expect. In addition, I was quite disappointed that the number of sources used did not change (and the References part is again a mess with repeated lines).

Incorporating the other comments, and probably other reviewers' comments, has improved the feel of the article, but the content has changed/extended only a bit. Therefore, I consider the above comments to be essential.

I have to repeat my conclusion from the previous review:

It is not appropriate to publish the article in this form; it should be expanded with the facts and procedures mentioned above and possibly supplemented with graphs to view the results obtained.

Author Response

To Reviewer 1

Unfortunately, I consider the submitted version of the article to be inconsistent in terms of formality and format.

The added text in section 2.1 is messy, and in the last paragraph in section 1 again. This situation is probably the result of tracking changes, but it is advisable to check the author's reviewed document before submitting it.

We have revised the section and also paid attention to the tracking changes.

A description of what Microsoft Teams is probably not the appropriate content for this type of article. Moreover, this information is not at all necessary for the set goals.

Although the reviewer is right on this concern; this information was added since requested by other reviewers.

The added content in section 2.4 is again chaotic; on page 7 again. The used form of change tracking saved into pdf format is very confusing. It is more appropriate to use docx format or color differentiation of changed parts without bubbles in the future.

We have included the in-depth analysis of the definition of a single index of mental distress in Appendix 1, to make the text less chaotic and more understandable, since this aspect does not concern the main objectives.

Statement on the incorporation of comments from the previous review round:

I agree with the definition of the two goals of the study in the revised paper. However, I still do not see the relationship between these goals. On what basis do the authors conclude that there is a relationship between these characteristics? Why did they decide to investigate the correlation between them? What should the finding of correlation bring? If at least one of these questions cannot be answered, the article in this form makes no sense, and it is appropriate to divide it into two. The common goal is partially touched by the definition of Mental Distress in section 2.4. Still, it is the GOAL of the article, and it is necessary to define it or its starting points or the idea of a solution in a clear form earlier.

We have better specified the rationale of the paper, with the close relationship between the two goals.

Several of my comments were related to the used bibliographical sources. Although I received several responses to the comments from the authors, they mainly did not satisfy me. In addition to the author team, other authors are undoubtedly dealing with related topics. An analysis of their results and approaches would enrich the article in Related work, which is not as represented in the article as I would expect. In addition, I was quite disappointed that the number of sources used did not change (and the References part is again a mess with repeated lines).

Although we had added some papers in the previous revisions, we have now expanded the current research on the topic (beyond the few papers by our group) and corrected the references.

Incorporating the other comments, and probably other reviewers' comments, has improved the feel of the article, but the content has changed/extended only a bit. Therefore, I consider the above comments to be essential.

Thanks for the suggestions, we have reviewed the work and integrated the required changes.

I have to repeat my conclusion from the previous review:

It is not appropriate to publish the article in this form; it should be expanded with the facts and procedures mentioned above and possibly supplemented with graphs to view the results obtained.

We regret that the revisions did not satisfy the reviewer, as we had tried to address his/her comments carefully. However, thanks to the re-examination of the previous concerns and these further comments, the paper has been expanded and now much improved (in our opinion).

Reviewer 2 Report

Dear Author,

The manuscript has been significantly revised. I think there is still a lot to be learned from this research. I look forward to your next research.

Author Response

To Reviewer 2

The manuscript has been significantly revised. I think there is still a lot to be learned from this research. I look forward to your next research.

Thank you for the positive evaluation of the paper.

Reviewer 3 Report

You made significant improvements to the article. Well done.

Author Response

Thank you for your opinion and for the advice given that made the manuscript more interesting.

Reviewer 4 Report

Dear authors,

The manuscript has gained quality after the changes made.

Congratulations!! the manuscript has gained solidity.

I just want to comment that the following comment has not been answered: "It would be convenient to indicate the % of non-response rate and the reason". 

Kind regards.

Author Response

To Reviewer 4

The manuscript has gained quality after the changes made.

Congratulations!! the manuscript has gained solidity.

I just want to comment that the following comment has not been answered: "It would be convenient to indicate the % of non-response rate and the reason". 

We apologize for not having answered this question earlier. We have added this data in the results section: 70% of the subjects contacted to participate in the survey did not respond. This can be due both to online media that can create a non-delivery (the email can end up in spam, the message cannot be read) and to personal reasons, such as boredom in filling in or unwillingness to give consent to use of data for research.

Round 3

Reviewer 1 Report

I thank the authors for the precise inclusion of comments. So, I think that in this form, the quality and readability of the article have improved significantly.

The addition of other sources and opinions of other authors on the issue, as well as the confirmation of the obtained results, also significantly contributed to the improvement of the content.

I would consider integrating the appendix into the text of the article. I think that the content of the appendix will not disrupt the flow of reading, and the reader can grasp the content better in the current form of the article.

In conclusion, I congratulate the authors for creating an interesting article.

Author Response

Thanks for your opinion, we have integrated the appendix in the statistical analysis section, as suggested.